# Condition dependent strategies of egg size variation in the Common Eider *Somateria mollissima*

**Thomas Kjær Christensen**⊙*, **Thorsten Johannes Skovbjerg Balsby**

Department of Bioscience, Aarhus University, Rønde, Denmark

* tk@bios.au.dk

**Data Availability Statement:** Data is owned by a third-party, A/S Øresundsbrokonsortiet, which approved the use of the data for this study. Data cannot be shared publicly because data is part of a comprehensive Impact Assessment study

## Abstract

We analysed intraclutch egg-size variation over the laying sequence in relation to clutch size, and the relation between clutch size and female body condition, in the Common Eider *Somateria mollissima* during an 8-year period. The aim was to assess if eiders adjusted egg size within the laying sequence depending on clutch sizes in response to body condition, as such an adjustment could have adaptive implications on reproductive success through a size advantage for the hatchlings. The analyses were performed on a population level; and then at the individual level using data from recaptured females that changed clutch size between years. Based on 1,099 clutches from 812 individual females, population clutch size averaged 4.13 eggs (range: 1–6), with 4- and 5-egg clutchesconstituting *c.*70% of all clutches, taking turns in being the most represented clutch size. Clutch size was positively related to female pre-laying body condition at both the population and individual levels. Egg size varied significantly within and between clutch sizes and changes were significantly related to the laying sequence. First eggs were significantly larger in 4-egg clutches and second eggs marginally smaller than in 5-egg clutches, a pattern also found among individual females changing clutch size between years. The relationship between female pre-laying body condition and clutch size, and the intraclutch egg-size pattern indicate that both clutch size and egg size are actively adapted to the pre-breeding body condition of the female. As egg size potentially optimise reproductive success through a size advantage in hatchlings, the observed pattern of intraclutch egg-size variation suggests that female eiders possesses a finely tuned conditional dependent mechanism that may optimize reproductive output in years were females are in suboptimal body condition for breeding.

## Introduction

Intraclutch egg-size variation has frequently been described in both altricial and precocial avian species [1–3]. In large long-lived precocial waterfowl species, such as geese and eiders, egg-size variation over the laying sequence generally follow a rather fixed curvilinear pattern, with increasing size from first to second and/or third egg and then a declining size over the following eggs in the laying sequence, with the final egg being the smallest [4–10]. With few

commissioned by the A/S Øresundsbrokonsortiet. Data are available upon request and acceptance from the A/S Øresundsbrokonsortiet (contact via kontakt@oresundsbron.com) for researchers who meet the criteria for access to confidential data. The authors of this study had no special access privileges.

**Funding:** The author(s) received no specific funding for this work.

**Competing interests:** The authors have declared that no competing interests exist.

exceptions, this pattern of intraclutch egg-size variation is consistently found among the most commonly laid clutch sizes within different eider populations, occurring almost independently of clutch size, female age and condition, and time of laying [3,4,7,11,12]. Parsons [13] suggested that the small size of the first egg was related to a lower physiological efficiency of females at the up-start of egg laying, while other studies support a physiological and hormonal, genetically based, explanation to the overall pattern of intraclutch egg-size variation [see 4]. Alternative but not mutually exclusive hypotheses, have related intraclutch egg-size variation to the depletion of resources in laying females with egg number, to incubation strategy, to facilitation of hatching synchrony and to differential investment in eggs with highest probability of hatching [8,10,11,14–22]. Despite the long-term focus on intraclutch egg-size variation, neither adaptive nor non-adaptive explanations have proved to fully explain the observed egg size patterns.

Although egg size in birds generally show high repeatability and heritability, and consequently is less flexible in individual birds than e.g., clutch size [cf. 23–25], egg size and intraclutch egg-size variation may have important implications for the reproductive success of individual females. In geese and eiders, wich lay small clutches of 4 to 6 (1–7) relatively large eggs, larger eggs within the clutch produce larger hatchlings that have a faster growth rate, higher pre-fledging survival and even a higher recruitment rate [3,26–31,but see 32]. However, despite the apparent benefit of making adaptive changes (e.g. laying larger eggs when laying a reduced clutch size or vice versa, or changing egg size according to female conditional state or in relation to environmental conditions), such adaptive intraspecific changes have not been substantially documented [see 5,33–35]. This is surprising, as even small proportional changes in egg size that would make equal egg size within clutches, is expected not to be costly to females already being physiologically upgraded for egg laying [cf. 11,24]. Likewise, avian life history characteristics suggests that adaptive patterns in both clutch and egg size should be more prevalent in species that store resources for reproduction in the pre-breeding period (capital breeders) compared to species depending on resources available during the laying and chick-rearing period (income breeders) [36]. Thus the apparent lack of flexibility in egg size even in capital breeding birds, suggests that adaptations to varying conditional states of females is mainly adjusted through changes in clutch size [cf. 24]. Indeed, many studies have shown a proximate linkage between female body condition and clutch size, as well as an increase in reproductive success with increasing clutch size [e.g., 14,23,37–39].

In a long-term study of breeding eiders in Denmark, records of egg size showed that intraclutch egg-size variation over the laying sequence followed the common curvilinear pattern as generally described in eiders [see 7,9]. However, we also observed that egg size in the most commonly laid clutches of 4 and 5 eggs showed a systematic difference with respect to the size of the first laid egg. Examining previous studies of eider clutch and egg size [cf. 7,9,10], we found a similar difference in other breeding populations, suggesting that eiders adaptively change the size of first laid egg depending on clutch size.

In the present study, we use data from a long-term survey of breeding eiders to test if the common eider females change egg size within the laying sequence depending on clutch size, which may have adaptive value in terms of reproductive success. Since clutch size in the eider is closely related to female body condition, we also included female body condition in analyses of clutch size and clutch size changes. We analyse data at the population level, as well as the individual level, based on ringed and recaptured individual females that changed clutch size within our study period of 8 years, and discuss our results in relation to the potential impact on reproductive success.

## Materials and methods

### Data collection

Data were collected annually during 1993–2000 at the island of Saltholm, Denmark (55˚55'N 12˚46'E). Access permits to the island was provided by the Saltholm Ejerlaug and the Danish Forest and Nature Agency. During the study period the island, covering 13.5 km$^2$, held an estimated breeding population of 4,000 to 6,000 eiders [40]. In all years, seven 45 m wide coast-to-coast transects, covering an area of c. 0.75 km$^2$, were searched thoroughly for nests towards the end of the incubation period in early May. Transects were distributed evenly 1 km apart following east-west oriented UTM grid lines. Attempts were made to capture all encountered incubating females, but only data from clutches where females were caught (assumed incubating on a fully laid clutch) were included in the present analyses. Two females sitting on clutches of 7 eggs were omitted as egg size and coloration strongly indicated egg dumping in both nests. We consider that the extremely low nest densities (average 0.65 nest per 45 x 45m sampling unit ~ 3.2 nest per ha [40]) relative to other studies generally precludes nest parasitism, which increases with nest density, being reported in colonies above 10–15 nest per ha [cf. 41]. However, we cannot reject that a few incomplete or parasitized nests, as well as some partially predated nests (in cases where no egg-shell remains was found to disqualify the record) may unintentionally have been included in the present data, and hence have induced some uncontrolled variation.

During 1993 to 2000, 1,099 incubating females were captured on nests (between 79 and 174 per year), ringed with standard steel leg bands and body mass (to nearest 5 gram) and tarsus length (to nearest 1 mm) were recorded. The sample included 812 individual females, of which 610 were caught once, 141 twice, 44 thrice, 12 four times, 3 five times and 2 six times.

Captures and ringing of eiders were conducted according to the practical and ethical guidelines for bird ringing and carried out under the institutional ringing license (no. 600) issued by the Ringing Centre at the Natural History Museum of Denmark.

The 1,099 clutches contained a total of 4,531 eggs. In all clutches, individual eggs were ordered in their laying sequence (first to sixth egg) based on individual degree of staining [4,8,34, see also 42], a method previously applied to eiders [23]. Egg length and width were measured to nearest 0.1 mm using callipers.

Egg volume (hereafter referred to as egg size) was subsequently calculated according to the formula: Volume (egg size) = length × π × bredth$^2$ × 0.00164 cm$^3$ [9,43]. In each year, female body mass prior to egg laying ($W_{start}$) was estimated for each individual female based on the difference between body mass at capture ($W_{capt}$) and the average year specific body mass at hatching ($W_{hatch}$). $W_{hatch}$ was obtained from regressions of body mass on the tarsus length (cubed) of all females captured with ducklings or hatching eggs in their nests. This calculation was done separately for all years, and hence accounts for annual differences in average population body condition (weight related to individual structural size). We used a daily weight loss of 20 gram during incubation and an incubation period of 28 days [cf. 44] to calculate individual body condition at start of egg-laying as

$$W_{start} = W_{capt} + \left(28 - \frac{\Delta W}{20}\right) \times 20 + (clutch\ mass)$$

where Δw is the difference between $W_{capt}$ and $W_{hatch}$ for a given structural size. Clutch mass was estimated setting average egg mass to 100 gram [cf. 45,46].

To ensure that clutch size differences were not responsible for differences in $W_{start}$-estimates, we tested relations between clutch sizes and observed $W_{hatch}$. The analysis suggested

that clutch size did not affect $W_{hatch}$ whereas tarsus[3] significantly related to $W_{hatch}$ (see S1 File).

In the analyses of intraclutch egg-size variation, we only included clutches of 2 to 6 eggs (98.5% of all clutches recorded) given the uncertainty associated with clutches of 1 (likely incomplete) and 7 eggs (potentially parasitized) (please see S1 Table for distribution of clutch sizes). When analysing individual egg-laying patterns in recaptured individuals, we only included 4 and 5-egg clutches, as these were numerically dominating (> 70% in all years).

## Statistics

**Clutch size.** We used general linear models (GLM) to test if clutch size differed between years and whether clutch size related to female body condition at the start of egg laying. To test whether clutch sizes of four or five dominated in different years we used a chi–square test [47].

**Clutch size and body condition for recaptured females.** For recaptured females, the likelihood of changing their reproductive strategy with regard to clutch size in a given year was tested by relating between-year changes in clutch size to between-year changes in body mass at the start of egg laying. We classified female reproductive strategy as either reducing, maintaining or increasing clutch size. To account for multiple contributions by some individuals (up to 4 recaptures in different years) we included individual females as a random factor in a mixed model. The likelihood of a change (increasing, maintaining or reducing) in clutch size as a response to differences in pre-laying body mass were tested by a generalized linear model with a multinomial distribution. We calculated least square pairwise differences for post hoc comparison of differences in egg sizes between clutch sizes. The proportion of significant pair-wise differences by far exceeded 5% expected by random, and hence did not require adjustment of significance threshold.

**Inter- and intraclutch egg-size variation.** We tested egg size differences within and between clutch sizes for first, second and third egg, separately using a mixed model including clutch size as a fixed effect and year as a random factor.

**Patterns of egg size variation in relation to laying order.** We tested the pattern of egg size differences over the laying order among clutch sizes using a repeated measure ANOVA, to detect if the pattern with laying order differed between different clutch sizes. The model included laying order, clutch size and year, and interactions between laying order and clutch size, and between year and laying order. In this model, we used clutch size and laying order as discrete variables, because this bypasses the need to specify the relation (linear or non-linear). The model used individual clutches as the repeated measure. Note that the relationship between laying order and egg size could only be compared for the number of eggs that corresponds to the smallest clutch size selected (e.g. the pattern for the first four eggs could only be tested in clutches with at least four eggs). We therefore conducted separate repeated measures ANOVA for the first 2 eggs, then on the first 3 eggs, then on the first 4 eggs and then on the first 5 eggs.

**Egg-laying patterns within individuals.** The recaptured females allowed us to test if the strategy for 4 and 5 egg clutches differed when individual variation was removed. We used paired t-tests to test if the egg size of first and second egg differed in 4 and 5 egg clutches. To account for changes in egg strategy we analysed females that changed clutch size from 4 to 5 eggs separately from females that changed clutch size from 5 to 4 eggs. Eight females, however, were recaptured three times and thus entered the data set for both changes from 4 to 5 and from 5 to 4 egg clutch sizes.

Assumptions regarding normal distribution and homoscedasticity of residuals were fulfilled for all tests requiring it. We used SAS 9.4 (SAS Institute, Cary, NC) for all analyses using Proc Mixed, Proc GLM, Proc Ttest, Proc Univariate, and Proc Freq.

## Results

### Clutch size

Average clutch size during 1993–2000 was 4.13 ± 0.03 SE (N = 1,099). Clutch size approached statistically significant variation between years, with the largest clutch sizes recorded in 1993 and 1996 and the smallest clutch size in 1997 (range: 3.97–4.30; General linear model: $F_{7,1091}$ = 1.99, p = 0.054, $R^2$ = 0.013)(Fig 1) (S2 Table).

In all years, clutches of 4 and 5 eggs were the most frequently occurring clutch sizes (annual average 35.9%, range: 24.5–47.3% and 37.8% range: 30.3–50.4% respectively). A significant (Chi-square test: $\chi_6^2$ = 20.2, p < 0.005) shift in dominance occurred with 4 egg clutches being more frequent in some years and 5 egg clutches in other years. On average total clutch volume of 4-egg clutches was 80.3% of the volume of 5-egg clutches.

Clutch size was significantly and positively related to female body condition at the start of egg-laying (General linear model: clutch size: $F_{1,\ 809}$ = 1382.53, p < 0.0001, slope = 0.006).

### Clutch size and body condition for recaptured females

The majority of the recaptured females that reduced clutch size between years, showed a decrease in body mass (72 lost weight and 18 gained weight). The majority of the recaptured females that increased clutch size between years, showed an increase in body mass (86 gained weight and 12 lost weight). Among the 108 females that did not change clutch size between years there was no distinct pattern with regard to changes in body mass (43 females lost weight and 65 gained weight). The changes in individual female body mass at start of incubation between years significantly affected changes in clutch size (increasing, maintaining or reducing clutch size)(Generalized linear mixed model $F_{1,\ 287}$ = 104.6, p < 0.0001, slope = 0.011). Hence, when females lost body mass relative to the previous breeding record, they were significantly more likely to reduce clutch size, as indicated by the model estimates, and vice versa.

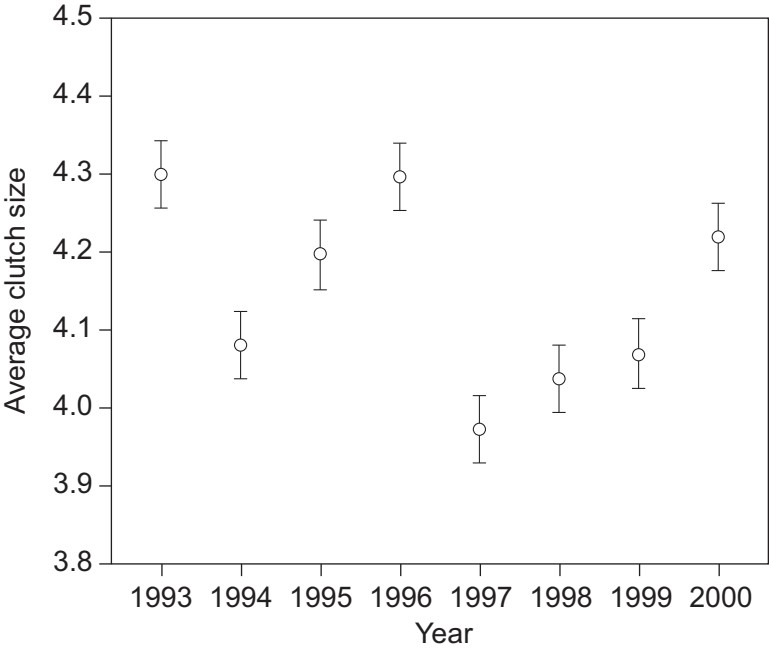

**Fig 1. Average clutch size (±SE: Standard error) for 1,099 clutches of eiders captured during incubation on the island of Saltholm during 1993–2000.**

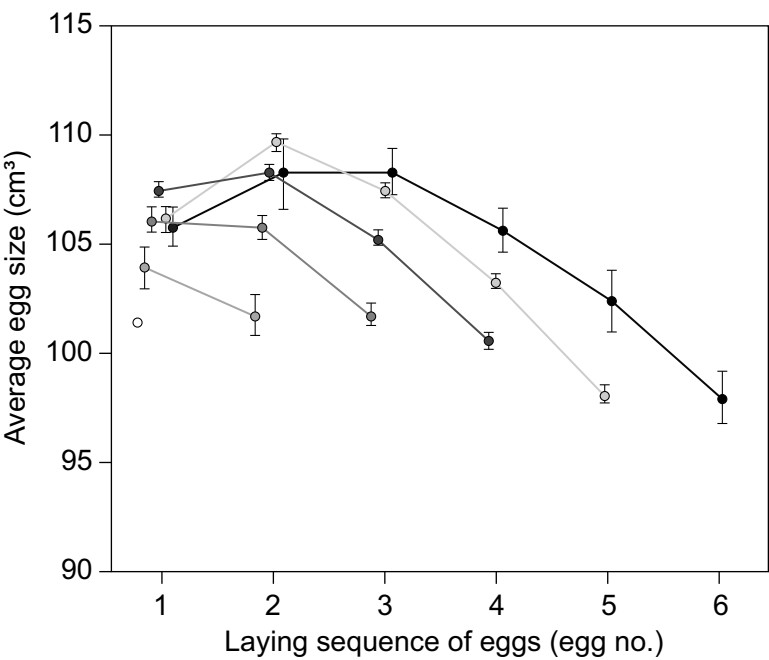

**Fig 2. Average egg size (±SE: Standard error) in eider clutches in relation to position in the laying sequence for females captured on nests with 1 to 6 eggs.**

## Inter- and intraclutch egg-size variation

Overall mean egg size in all clutches (N = 4,531) was 103.1 ± 0.1 cm³. Excluding extreme outliers of one dwarf (20.1 cm³) and one extremely large egg (176.7 cm³), egg size range was 79.4–159.5 cm³ (n = 4,529) and the smallest egg was 49% the size of the largest egg in the population of measurements.

Egg size showed a consistent pattern with laying order across different clutch sizes (Fig 2), and the size of the first, second and third egg differed significantly with clutch size (Table 1). The difference in the size of the first egg was related to a significantly larger egg in 4-egg clutches compared to all other clutch sizes (Fig 2, S3 Table). The second egg in the laying sequence was significantly larger in 5-egg clutches compared to 2- and 3-egg clutches, almost significantly larger (p = 0.054) than in 4-egg clutches, but did not differ from the second egg in 6-egg clutches (Fig 2) (S3 Table). The size of the third egg in 4-egg clutches was significantly larger than in 3-egg clutches, but significantly smaller than in 5- and 6-egg clutches.

## Patterns of egg size in relation to laying order

Egg size was significantly affected by laying order in clutch sizes of 2–6 eggs (Table 2). More importantly, clutch size significantly affected the size of eggs within the laying sequence as

**Table 1. Test of differences in egg size of first, second and third eggs across clutches of 2–6 eggs.**

| Egg size | df | F | p | N |
|---|---|---|---|---|
| First egg | 5, 804 | 5.49 | <**0.001** | **817** |
| Second egg | 4, 796 | 13.87 | <**0.001** | **808** |
| Third egg | 3, 752 | 17.22 | <**0.001** | **764** |

Data were analyzed with a mixed model with year as a random factor and only included first records of recaptured females. Bold font indicates statistically differences in egg size.

**Table 2. Test (repeated measures ANOVA) of egg size changes with clutch size, year and laying order, and the interaction terms for laying order for a stepwise increase in clutch size (ex: Egg 1 and 2 includes all clutch sizes larger than one egg (2 to 6 egg clutches); Egg 1 to 5 includes clutches with 5 or more eggs (5 and 6 egg clutches)).**

| | Egg 1 and 2 ($N_{clutch}$ = 1,087) | | | Egg 1 to 3 ($N_{clutch}$ = 1,030) | | | Egg 1 to 4 ($N_{clutch}$ = 844) | | | Egg 1 to 5 ($N_{clutch}$ = 450) | | | Egg 1 to 6 ($N_{clutch}$ = 35) | | |
|---|---|---|---|---|---|---|---|---|---|---|---|---|---|---|---|
| | df | F | p | df | F | p | Df | F | P | df | F | p | df | F | p |
| clutch size | 4, 1073 | 10.05 | **<0.0001** | 4, 1017 | 11.86 | **<0.0001** | 3, 832 | 4.87 | **0.008** | 2, 439 | 1.32 | 0.251 | | | |
| year | 7, 1073 | 0.99 | 0.437 | 7, 1017 | 1.24 | 0.276 | 7, 832 | 1.00 | 0.430 | 7, 439 | 0.66 | 0.705 | 6, 26 | 0.83 | 0.554 |
| order | 1, 1073 | 16.42 | **0.024** | 2, 2034 | 21.03 | **<0.0001** | 3, 2496 | 59.16 | **<0.0001** | 4, 1756 | 49.73 | **<0.0001** | 5, 130 | 13.70 | **<0.0001** |
| order*clutch size | 4, 1073 | 16.14 | **<0.0001** | 8, 2034 | 18.63 | **<0.0001** | 9, 2496 | 16.78 | **<0.0001** | 8, 1756 | 4.49 | **0.001** | | | |
| order*year | 7, 1073 | 1.71 | 0.102 | 14, 2034 | 0.96 | 0.491 | 21, 2496 | 0.97 | 0.495 | 28, 1756 | 0.99 | 0.481 | 30, 130 | 0.95 | 0.552 |

Bold fonts indicate statistically significant differences.

indicated by the significant interactions between laying order and clutch size for clutch sizes between 2 and 6 eggs (Table 2).

## Egg laying patterns within individuals

The difference in the intraclutch egg-size pattern of females laying 4 and 5 egg clutches in the general analyses (cf. Fig 2), was also found in recaptured females that changed their clutch size from 4 to 5 eggs (N = 35)(Fig 3A) and from 5 to 4 eggs (N = 43)(Fig 3B). When laying a 4-egg clutch, the size of first and second eggs were comparable in size (Paired t-test $t_{34}$, p = 0.839), but there was a marked size difference (Paired t-test $t_{34}$ = 3.78, p = 0.0006) when subsequently laying a 5-egg clutch, relating to a reduced size in the first egg and a slight increase in size of the second egg. For females changing clutch size from 5 to 4 eggs, the opposite change in the size of the first and second egg was evident (Paired t-test: 5-egg clutch: $t_{42}$ = 4.432, p<0.001; 4-egg clutch: $t_{42}$ = 0.07, p = 0.943t). Hence, individual females laying a 4-egg clutch increased the size of the first egg while decreasing the relative size of the second egg compared to years when laying a 5-egg clutch and vice versa. A similar, but less clear, trend was observed among females making two changes, either 4-5-4 eggs or 5-4-5 eggs, although the samples sizes were very small (N = 5 and 6 respectively) and hence subject to differences between individuals (Fig 4).

## Discussion

Our results confirmed a general intraclutch increase in egg size from the first to the second egg, with declining size in subsequent eggs, as commonly described in eiders and in several other Anatidae species [4,7,9,48,49]. However, we also found a consistent pattern of intra-clutch egg-size variation depending on clutch size for the most commonly laid clutches of 4 and 5 eggs. In 4-egg clutches, the first egg was significantly larger and the second egg almost significantly smaller than in 5-egg clutches. This population level pattern in egg-size with clutch size was also found among individual females when they both increased and decreased their clutch size between years. As clutch size was significantly related to female body condition, our results suggest that intraclutch egg-size variation may be an adaptive strategy adopted by individual females in response to their own pre-laying body condition.

## Clutch size variation

Female eider body condition prior to breeding is widely acknowledged as a major determinant of clutch size [23,50,51]. However, clutch size is also affected by female age and timing of

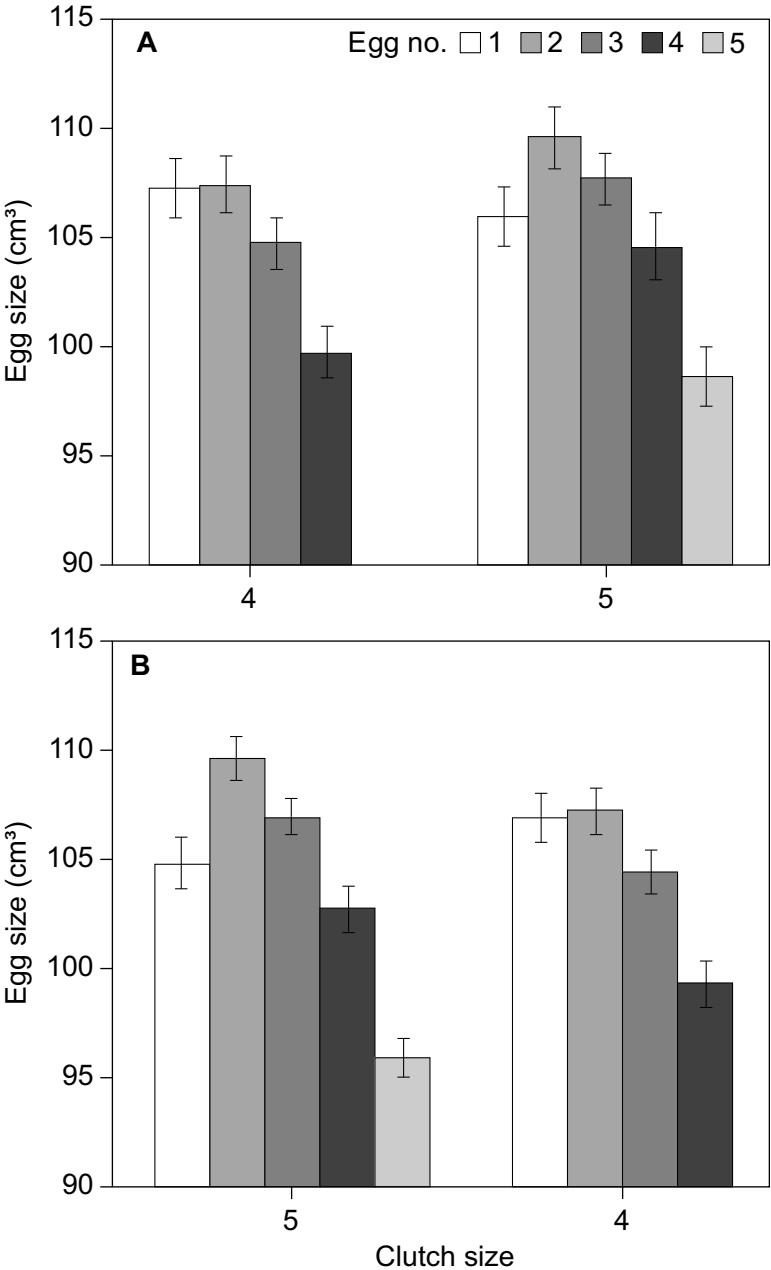

**Fig 3.** Average egg size (±SE: Standard error) for consecutive eggs in the laying sequence for individual female eiders that changed clutch size from A) 4 to 5 eggs (N = 35) and B) 5 to 4 eggs (N = 46).

laying [52,53], which often are interrelated, with young birds laying later than older birds [54]. In this study, nest surveys were always performed in the early hatching period, and annual average laying date varied by only five days throughout the study period [40], suggesting no effect of year from the timing of laying. Likewise, restricting the analysis to females caught on nests minimised the possible inclusion of late breeding young birds [cf. 55], as well as potential re-nesters. We therefore believe that the variation in average clutch size between years mainly reflects varying environmental conditions affecting females during winter and the pre-breeding period.

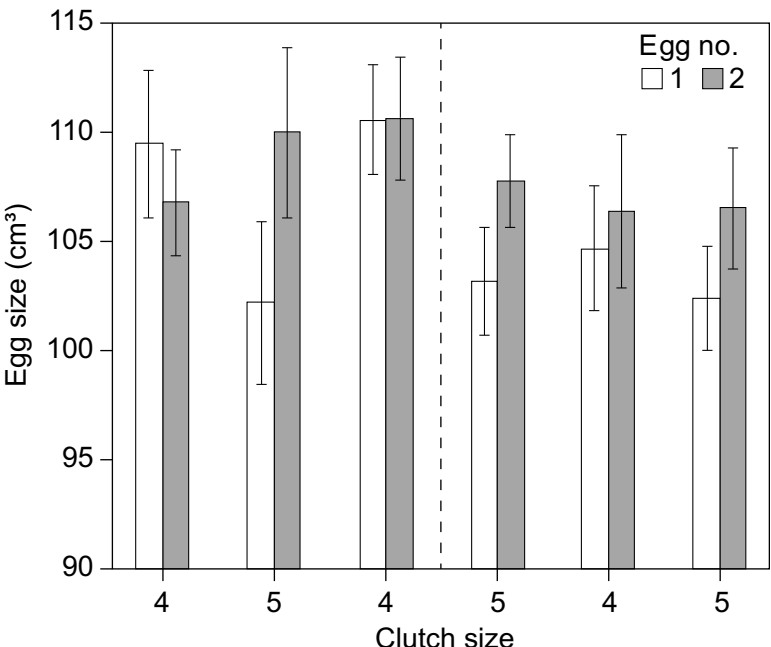

**Fig 4. Average egg size (±SE: Standard error)) for egg number 1 and 2 in the laying sequence for individual female eiders that made two clutch size changes: From 4 to 5 to 4 eggs (N = 5) and from 5 to 4 to 5 eggs (N = 6) between years.**

## Intraclutch egg-size variation

Overall, the present data reflect the general pattern of intraclutch egg-size variation reported in many waterfowl species as well as in other eider populations [e.g., 7,9,11]. However, we show that egg size was significantly related to laying order (in 2–6 egg clutches), and, for the first time, that the pattern of egg size variation with laying order was significantly affected by clutch size (in 2–6 egg clutches). This suggests that eider females produce eggs with a specific predetermined pattern of intraclutch size variation depending on ultimate clutch size. Since this pattern alternated in individual females when both increasing and decreasing clutch size between years (in relation to associated pre-laying body mass), our results strongly support the interpretation that eider females actively adapt clutch and egg size as part of their reproductive strategy, following some pre-programmed clutch/egg size "formats".

Overall, the absolute difference in the size of the first and second eggs in 4 and 5-egg clutches was ~1.4 cm$^3$, corresponding to approx. 1.2% of the average egg size of the first two eggs. This relatively small change in the first laid egg has not been commented in previous studies, even though matching patterns showing a larger first egg in 4-egg clutches compared to 5-egg clutches, have been reported in other eider populations (Fig 7 in [7]; Table 1 in [9], see also [10]).

## Implications of changing egg size

Egg size is recognised as a major determinant of hatchling success, and increases in egg size directly affects the size of ducklings within and between clutches, and hence may have significant fitness value [56,57,]. Likewise, reproductive success, in terms of number of ducklings leaving the nests and higher duckling survival, increases with clutch size [58–60]. Hence, increasing the size of the first laid egg in 4-egg clutches, has the potential to partly compensate

for overall reduced reproductive success compared to a higher success in 5 and 6 egg clutches. Thus, when laying a 4-egg clutch, females produce two large eggs, which are almost comparable in size with the two largest eggs in a 6-egg clutch and comparable to the second largest egg in a 5-egg clutch (cf. Fig 2). Hence, instead of producing only one large egg, if the size of the first egg was similar to that of 5 and 6 egg clutches, females laying 4-egg clutches lay two large eggs and hatch two relatively large ducklings, with the likely associated fitness benefits. In geese and large waterfowl species, greater size at hatching is reported to benefit individuals by a stronger competitive ability for optimal brooding, more efficient foraging skills, stronger and quicker responses to predator warning in female and/or a more efficient physiological/immunological state. Such traits likely contribute to an increase in survival [cf. 28,61].

The advantage of laying two relatively large eggs may however, not only be restricted to the benefit of larger size in individual ducklings within a clutch of 4 eggs. As eiders frequently engage in brood amalgamation, forming so-called crèches [62,63], larger ducklings may benefit from their size advantage at the multi-brood/cohort level when grouped with other ducklings from one or more broods, or abandoned by the female into crèches [but see 64]. Analysed in relation to the expected size variation in a duckling cohort based on egg sizes, the present data (cf. Fig 2) show that among the largest 20% of all ducklings in an average cohort, almost half (47.6%) originate from 4-egg clutches (from the first and second egg), 48.6% originate from 5-egg clutches (from the second and third egg), and 3.8% from 6 egg clutches (from the second and third egg). Without the recorded size increase in the first egg (compared to the size of first egg of 5- and 6-egg clutches), 4-egg clutches would only represent 31.2% of the largest ducklings in a cohort, while 63.8% and 4.9% will be from 5- and 6-egg clutches respectively. Hence, the egg size pattern of 4-egg females may be of adaptive advantage in terms of reproductive success, as the relative duckling size within cohorts determine survival and recruitment [26], and by allowing females in a suboptimal body condition to reduce clutch size and/or abandon ducklings without impairing their survival [cf. 65].

## Compliance with general hypotheses

Most studies of intraclutch egg size in eiders and other precocial waterfowl species show variation consistent with the adaptive resource distribution hypothesis. This hypothesis states that females allocate most resources to eggs with a higher probability of survival, which reflects their position in the laying sequence [4,8,9,11,66, see also 36]. In eiders, the highest energy investment, assessed here by larger egg size, is allocated to the first 3 eggs in the laying sequence. Of these, pre-incubation failure (mainly predation) of unattended first laid eggs is much higher than for subsequent eggs [67,68], whereas higher survival in subsequent eggs is attributed to increased nest attendance and incubation initiation, which start after the laying of the second or third eggs [10,67,68]. Hence, increasing the size of the first laid egg when laying 4 eggs, as found in the present study, seems a highly risky investment, if not accompanied by an earlier nest attendance. We did not investigate the timing of nest attendance and incubation initiation in the present study, but Hanssen et al. [10] showed that eiders laying 4-egg clutches start incubation on average one day earlier in the laying sequence than those laying 5-egg clutches. They also showed that an earlier start to incubation was related to poor female body condition, but they did not relate advanced incubation to a larger size of the first laid egg in 4-egg clutches, even though their data showed such a pattern (see Hanssen et al. [10]). In snow geese, Williams et al. [32] reported that larger first laid eggs had a higher survival prior to incubation start than smaller first laid eggs. They did not consider timing of the start of incubation in their study, but their results also indicated that female snow geese laying large first eggs in a clutch were capable of increasing survival of these eggs.

In theory, adaptive changes in egg size when laying a suboptimal clutch size, should occur in eggs of intermediate size and not in the smallest or largest eggs. Hence, in both 4-, 5- (and 6-)egg clutches, expected candidates for adaptive changes in size would be egg number one and three. The present analyses showed a size increase in the relatively large first laid egg, in line with our prediction, but this result somehow conflicts with the resource distribution hypothesis, as first laid eggs generally are reported to have a high failure rate (see above).

Although an earlier incubation start may represent an adaptation to increase first egg success in 4-egg clutches [cf. 10], a size increase in the third laid egg would have been more in line with the resource distribution hypothesis, with largest investment in the two central eggs. However, as previously stated, increasing the size of the third egg would probably postpone incubation initiation and further increase pre-incubation failure (predation risk) of first laid eggs, as incubation initiation may not commence prior to laying the largest egg within a clutch, without inducing a hatching or developmental asynchrony in the embryos [cf. 69]. Alternatively, the laying of two large consecutive first eggs of comparable size may facilitate an earlier incubation initiation, as females may increase nest attendance or start incubation after laying the first of these similar sized eggs, without jeopardizing synchronous development and hatching. If so, the laying of similar sized first and second egg in 4-egg clutches may explain the one-day earlier incubation initiation in these clutch sizes [cf. 10]. Likewise, similar egg size in 6 egg clutches, where the second and third egg are of comparable size, may reflect an adaptation to advance incubation start from the third to the second egg. Although no data exist on the time of incubation start in 6 egg clutches, and sufficient data may be hard to obtain, an advanced incubation start, comparable to that for 5-egg clutches, could explain how reproductive success, expressed as successfully fledged ducklings, generally increases with clutch size [58,59].

In conclusion, our study shows that individual female eiders have a specific within-clutch egg-size strategy, dependent on clutch size. Based on sequential histories of recaptured females, where individual females laid larger first eggs in a 4-egg clutch when in poor condition, and laid smaller first eggs in a 5-egg clutch when in better condition, we argue that this change is a functional adaptation, and that this adaptation may increase reproductive output, as the size increase in first egg results in two ducklings (rather than one) that potentially benefit from large size, in both single or amalgamated broods. As other studies have reported advanced incubation start in females in poor condition laying 4-egg clutches, we hypothesise that the increased investment in first laid eggs, which normally suffer from a high predation rate is, at least partly, compensated by an earlier start of nest attendance and incubation. Hence, our results suggest that eiders possess a finely tuned egg and clutch size strategy, by which individuals can adjust current reproduction in relation to their pre-laying body condition.

## Supporting information

**S1 File. Test for the effect of clutch size on $w_{hatch}$.**
(DOCX)

**S1 Table. Distribution of clutch sizes.**
(DOCX)

**S2 Table. Least square mean estimates (LS means) in mean clutch size for year and test of post hoc pairwise differences.**
(DOCX)

**S3 Table. Test of pairwise differences in egg volume between clutch sizes (first to third egg).** Post hoc pairwise tests were estimated with least square mean.
(DOCX)

## Acknowledgments

We are grateful to all the people who helped in catching eiders and to measure egg size during all years, especially Jens Peder Hounisen, Ebbe B. Hansen and the late Henning Noer. Niels Adamsen is sincerely thanked for supporting the logistics on the island of Saltholm, and the Saltholm Ejerlaug and Danish Forest and Nature Agency for providing permission to access the island. Data collection was part of a general investigation commissioned by the A/S Øresundsbrokonsortiet, who are thanked for permission to use data. Finally, Anthony D. Fox and Kevin K. Clausen are thanked for commenting on an earlier version of this manuscript. We would like to thank Pauline Toni for very constructive comments on previous versions of the manuscript.

## Author Contributions

**Conceptualization:** Thomas Kjær Christensen.

**Data curation:** Thomas Kjær Christensen.

**Formal analysis:** Thorsten Johannes Skovbjerg Balsby.

**Writing – original draft:** Thomas Kjær Christensen.

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
