## [Decision Letter · Decision Letter 0]

27 Feb 2020

PONE-D-19-33049

Condition dependent strategies of egg size variation in the Common Eider Somateria mollissima

PLOS ONE

Dear Mr. Christensen,

Thank you for submitting your manuscript to PLOS ONE. After careful consideration, we feel that it has merit but does not fully meet PLOS ONE’s publication criteria as it currently stands. Therefore, we invite you to submit a revised version of the manuscript that addresses the points raised during the review process.

Both reviewers appreciated the quality and interest of the manuscript, however, they both raised concerns about the clarity of the analysis  and the structure of the manuscript. Please address their comments carefully.

We would appreciate receiving your revised manuscript by Apr 12 2020 11:59PM. To enhance the reproducibility of your results, we recommend that if applicable you deposit your laboratory protocols in protocols.io, where a protocol can be assigned its own identifier (DOI) such that it can be cited independently in the future. For instructions see: http://journals.plos.org/plosone/s/submission-guidelines#loc-laboratory-protocols

We look forward to receiving your revised manuscript.

Kind regards,

Julien Martin

Academic Editor

PLOS ONE

Journal Requirements:

Reviewers' comments:

Reviewer's Responses to Questions

**Comments to the Author**

1. Is the manuscript technically sound, and do the data support the conclusions?

Reviewer #1: Partly

Reviewer #2: Yes

2. Has the statistical analysis been performed appropriately and rigorously? 

Reviewer #1: I Don't Know

Reviewer #2: Yes

3. Have the authors made all data underlying the findings in their manuscript fully available?

Reviewer #1: No

Reviewer #2: No

4. Is the manuscript presented in an intelligible fashion and written in standard English?

Reviewer #1: Yes

Reviewer #2: Yes

5. Review Comments to the Author

Reviewer #1: This article tests if common eider females adaptively adjust within-clutch egg size depending on clutch size, laying order and their own body condition. The authors use long-term, longitudinal data on a population of common eiders to analyse how variation in clutch size depends on maternal body condition; and how variation in egg size depends on clutch size and laying order, at the population and the individual levels.

The study has potential as the results are of interest, and are based on a strong dataset. The analyses at both the population and the individual levels bring very important information on egg size variation patterns. The study fails, however, to address the hypothesis the authors wish to test: that variation in within-clutch egg size patterns is an adaptive maternal adjustment. The authors analyse the direct relationship between maternal condition and clutch size, and between clutch size, laying sequence and egg size, but do not test the effect of within-clutch egg size variation patterns on any fitness trait (reproductive success in terms of number of hatchlings or offspring survival, or survival of the first egg). They instead rely on literature to make that link. Although literature on birds does state that egg size is related to maternal reproductive success, it has not, from the references listed, been conclusively tested in this population or species, and is not tested in this article. Moreover, laying sequence has been found to be as strong a predictor of hatching success and hatchling survival as egg size, according to some references cited in the manuscript (e.g. ref. 11). While it seems fair to interpret and discuss the results of this study in the larger context of reproductive tactics and adaptive adjustment framework, the aim and hypothesis should be reformulated and nuanced to focus more on and better fit with what the analyses actually test. If data are available to investigate the adaptive value of within-clutch variation (for example by testing survival probability of first eggs in 4- and 5-egg clutches), that should be done.

The methods section needs more work. The data collection part is sufficiently detailed, but the statistics part requires attention. Some of the results presented (clutch size, L202-206 and L216-218) are from analyses that are not described in the methods. Generally, some paragraphs are unclear and need reformulation or more details (e.g. L140-143, or the absence of details concerning linear model selection methods, linear model conformation to assumptions when applicable (homoscedasticity, normality)).

The structure of the manuscript is also at some place surprising, and overall inconsistent. The results are presented in a different order from that in the methods. The same goes for the discussion. The methods section describes an analysis of variation in first, second and third egg sizes across different sizes of clutch (L172-173) in very little details, an analysis of within-clutch egg size variation in relation to laying sequence and clutch size (L174-182), a third analysis of within-individual clutch size variation (L183-191), and a last analysis of within-clutch egg size variation for females with repeated measurements that changed clutch size (L191-194). The results section first presents descriptive results about clutch size and results from a linear model that seem to test the effect of year and female condition on clutch size, but that is not described in the methods section (L201-218). The section then presents descriptive data and results from the analysis body mass and clutch size variation between years for females captured several times (L221-232). Then are presented descriptive data on egg size (L235-239) and results from the linear model testing variation in egg size within clutches, in relation to laying order (L240-271). Finally, the authors present results of within-clutch egg variation according to clutch size for females that changed clutch size in subsequent years (L275-296).

There is also a lack of consistency in terminology (to distinguish tactic from strategy, see Gross 1996, in TREE; egg size and egg volume are used inter-changeably, as are body mass and body condition) and in the way the results are presented (sometimes F values, sometimes effect sizes but referred to as “slope” (L218) or “estimate” (L230), sometimes reporting the R² value of the model (L205)). It would be more informative to present the effect of the variables tested with linear models by giving the effect size (β) and standard error (or any error assessment) rather than F values.

Finally, there are throughout the manuscript several spelling, grammar and punctuation mistakes that need attention.

Detailed comments (with line numbers)

L22-23: the sentence needs to be reformulated: the authors looked at the effect of female body condition on clutch size, and then at the effect of laying sequence and clutch size on egg size, but did not analyse the direct relationship between female body condition and egg size.

L27-28: please reformulate, as “data from recaptured females…” only refers to the individual level of the analyses

L61: please remove the commas: “Alternative but not mutually exclusive hypotheses have related…”

L66: “have proved to fully explain the observed…”

L74-75: what about literature that found a trade-off between size and number of offspring? (even in Anseriformes e.g. Christians 2000 Trade-offs between egg size and number in waterfowl… or Figuerola and Green 2006 A comparative study of egg mass and clutch size in the Anseriformes)

L77-80: sentence unclear, needs reformulation

L86: no need for a comma: “…females is mainly adjusted…”

L86-88: these studies do not show that larger clutches, laid by females in good condition, have a higher reproductive success than small clutches

L89: “records of egg size”

L89-93: are these not results from the analyses described in the paper? It seems so, thus it is surprising to find them presented in the introduction.

L93: please specify the direction of the difference.

L94-96: please provide references

L99: “depending on clutch size”

L100: should be reformulated: the paper analyses the effect of body condition on clutch size, but body condition is not included in the main analysis of the effect of clutch size and laying sequence on egg size.

L108: it seems that all data were collected between 1993 and 2000, so it does not seem relevant to specify “Data on clutch size and egg size”

L109-110: it sounds weird to describe past density in a present tense; this sentence should be reformulated to indicate it was the density estimated at the time the data was collected.

L127: verb missing: “tarsus length […] were recorded”

L135: it is not very clear what “related to the laying sequence” means (numbered? ordered?), this part should be reformulated

L138: the fact that egg size in the MS will refer to egg volume should be introduced here, not L235.

L139 and throughout: please be consistent between body mass and body condition

L139-143: this is unclear, please reformulate

L149-150: it is not clear if body mass refers to actual measured mass or the calculation of body condition

L150: it is unclear if “annual body mass at hatching” refers to maternal or hatchlings body mass

L155-164: this paragraph should be moved to supplementary, and referred to in the paragraph detailing the calculation of female body condition in terms of how the potential effect of such and such variables on body condition was considered and controlled for. Moreover, details are missing concerning the structure of the model ran and the model selection process: it is not clear if the values presented are from a single model. If so, F values of the variables when not involved in the interaction (clutch size and tarsus3) are irrelevant as in the case of an interaction, the effect of the variables involved should not be interpreted separately from the effect of the interaction.

L157: “in which hatching […] was recorded”

L158: please add that Wstart is the response variable for clarity

L165-169: it would be informative to have the number of clutches of each size somewhere (could be in supplementary), to have an idea of the sample sizes of each clutch size

L172-182: this paragraph is not very clear: does the first sentence refer to one analysis (LMM), then L174-182 to another one (ANOVAs)? Do the authors mean they ran repeated measures ANOVAs as post-hoc tests to the linear mixed model? Does individual nest (L178) refer to each egg within a same clutch or an actual nest (and then if a same nest is used several years is it considered as a different nest each year)?

L183: “their reproductive strategy”

L183-191: this paragraph needs to be reformulated for more clarity, it seems that some sentences repeat the same information, and the type of analysis ran should be stated earlier.

L184: it is not clear if a season correspond to a year, or if inter-seasonal changes correspond to yearly changes

L188: it is unclear if individual refer to female ID. Please make sure to distinguish the significance of words that are used for different purpose (individual nest L178)

L191: please be consistent with the terminology used: egg size or egg volume (valid throughout the manuscript)

L205: why present the R² here but not when presenting results from other models?

L213: it is unclear what the authors mean by “shift in dominance”. Is it that in some years there were more 4-egg clutches while in other years 5-egg clutches were dominant?

L218: there are a comma and a space missing between “0.0001” and “slope = 0.006”. Also please be consistent with “,” and “.” when writing numbers.

L227-232: unclear: are females that lost weight more likely to reduce clutch size than females that gained weight, or than when they gain weight? Providing a figure would help represent this result

L241-242: it is unclear if this refers to a difference between first eggs from different clutch sizes, second eggs from different clutch sizes, and third eggs from different clutch sizes, or if they refer to differences between first, second and third eggs across clutch sizes. Does it show the same results as Fig. 2 but in a different way?

L253-257: please state as a bottom note to the table that bold means significant effects and add sample sizes in the title. “data were analyzed with a mixed model”

L261-265: these sentences are a bit stating the obvious… Please reformulate to indicate the direction of the effects.

L267-271: please state as a bottom note to the table that bold means significant effects and add sample sizes in the title. “data were analyzed with a mixed model”

L275-277: please reformulate the sentence to make it clearer

L279-280: please state the direction of the difference

L281-282: please describe the relation documented

L307: “and the second egg marginally smaller than in 5-egg clutches.”

L310-312: the indirect relationship between female body condition and within-clutch egg size variation does not suffice to conclude that egg-size variation patterns have an adaptive value. Plus, laying sequence has been suggested to be as strong a predictor of hatching and nestling survival as egg size. Without testing the effect of such patterns on a measure of fitness, within-clutch egg size variation as a reproductive tactic can at best be interpreted as a hypothetical explanation, hence should be discussed later in the discussion.

L325: is the environment really stochastic?

L325: ‘affecting conditions for breeding females’ is unclear: do the authors refer to the impact of environment on female body condition?

L331-333: please reformulate: the link between egg size and laying order has already been documented, as cited in the introduction (L50-54, L90-91). “for the first time” only applies to the link between within-clutch egg size variation, laying order and clutch size (second part of the sentence)

L343-345: the larger first egg in 4-egg than in 5-egg clutches is significant only in ref 9

L340-345: it is unclear what point this paragraph is making

L352: please remove the parentheses

L363: “predator warning in females”

L364-365: please reformulate, for example: “Such traits likely contribute to an increase in survival”

L366-367: please rephrase this sentence to make it clearer

L371: the formulation “in such a cohort ‘size-scenario’” is unclear

L373-378: are the two percentage distributions statistically different, and statistically different from an expected one based on probabilities? (I’m just curious of the significance of these numbers)

L380: please remove the parentheses

L380-382: that is a really interesting point

L382: “abandon ducklings without impairing their survival”

L387-390: please break the sentence down

L405-406: good point, one potential measure of reproductive success that could be investigated in the present study to test the adaptive value of within-clutch egg size variation, since it was the aim of the study.

L419-415: this paragraph repeats information given in L387-397.

L441-443: well, the study did not test that an increase in first egg size resulted in an increase in the number of large ducklings for 4-egg clutches.

L443-446: this is a hypothesis and not a result of this study, hence this sentence should be more nuanced.

L448-449: there is little information on the link between female body condition and environment in the manuscript, plus common eiders are capital breeders, hence this seems an odd conclusion to the paper.

Fig. 3: presenting these results following the same format as Fig. 2 would make comparisons easier to make for the reader

Reviewer #2: The goal of this paper is to assess whether eiders adjust egg size within the laying sequence depending on clutch size, with the ultimate goal of understanding whether egg size adjustment is adaptive (via optimizing hatching size). This work utilizes an impressive dataset and provides comprehensive evidence that eiders lay first eggs are larger and second eggs smaller in clutches of 4 eggs compared to 5 eggs. I find the paper generally sound and most of my comments are minor and focus on clarifying potential points of confusion.

Line 35 and elsewhere - Rather than wording like “marginally insignificant”, it might be more clear to just say “showed x trend” or “tended to..” just to make the direction of the almost significant relationship more clear.

I found the estimates of female pre-laying body condition quite difficult to follow. Were individual females weighed both at capture and at hatching? Otherwise where does the annual mean mass at hatching come from? Although there are references to mass lost per day of incubation etc, there is no mention as to whether this formula was ad hoc or if there is a precedent for it. It seems to make sense but it takes the reader a while to understand so if possible I’d suggest a bit more detail as to how this particular formula was derived, as well as where the mean mass at hatching comes from.

The wording (particularly in the abstract) about this ‘adaptive’ strategy can come across a bit strong, given the way body condition etc were calculated and without a more comprehensive analysis of female energetics during egg laying/incubation and how it corresponds to increased survival of both the individual female and her chicks. I think the discussion is more moderate in tone so seems appropriate, but I think it would be prudent to adopt that language in the Abstract, rather than stating that this represents a “finely tuned conditional dependent mechanisms that enable females in a suboptimal condition to optimize reproductive output”, which seems a bit over-stated.

6. PLOS authors have the option to publish the peer review history of their article (what does this mean?). If published, this will include your full peer review and any attached files.

Reviewer #1: Yes: Pauline Toni

Reviewer #2: No

---

## [Author Response · Author response to Decision Letter 0]

14 Apr 2020

All comments to editor and reviewers are included in the cover letter: Response to reviewers.

---

## [Decision Letter · Decision Letter 1]

8 May 2020

PONE-D-19-33049R1

Condition dependent strategies of egg size variation in the Common Eider Somateria mollissima

PLOS ONE

Dear Mr. Christensen,

Thank you for submitting your manuscript to PLOS ONE. After careful consideration, we feel that it has merit but does not fully meet PLOS ONE’s publication criteria as it currently stands. Therefore, we invite you to submit a revised version of the manuscript that addresses the points raised during the review process.

We would appreciate receiving your revised manuscript by Jun 22 2020 11:59PM. To enhance the reproducibility of your results, we recommend that if applicable you deposit your laboratory protocols in protocols.io, where a protocol can be assigned its own identifier (DOI) such that it can be cited independently in the future. For instructions see: http://journals.plos.org/plosone/s/submission-guidelines#loc-laboratory-protocols

We look forward to receiving your revised manuscript.

Kind regards,

Julien Martin

Academic Editor

PLOS ONE

Additional Editor Comments (if provided):

I found the manuscript much improved and clearer. I sent it back to only 1 of the 2 reviewers who has some more suggestions and minor comments. Please carefully consider them. I probably not send the next revision back to reviewers and accept it if I am happy with the edits.

Reviewers' comments:

Reviewer's Responses to Questions

**Comments to the Author**

1. If the authors have adequately addressed your comments raised in a previous round of review and you feel that this manuscript is now acceptable for publication, you may indicate that here to bypass the “Comments to the Author” section, enter your conflict of interest statement in the “Confidential to Editor” section, and submit your "Accept" recommendation.

Reviewer #1: All comments have been addressed

2. Is the manuscript technically sound, and do the data support the conclusions?

Reviewer #1: Yes

3. Has the statistical analysis been performed appropriately and rigorously? 

Reviewer #1: Yes

4. Have the authors made all data underlying the findings in their manuscript fully available?

Reviewer #1: No

5. Is the manuscript presented in an intelligible fashion and written in standard English?

Reviewer #1: Yes

6. Review Comments to the Author

Reviewer #1: This manuscript aims at testing how intra-clutch egg size varies with laying sequence depending on clutch size, and makes the link between such variation and maternal body condition, in common eiders. The authors use long-term, longitudinal data to answer these questions at both the population and the individual level. The study is of interest, as the results are sound and based on an impressively large dataset. The analyses at the population and individual levels bring important insights on egg size variation patterns. The paper offers an interesting discussion and interpretation of the results in the larger framework of female reproductive strategies.

This is the second time I review this manuscript (revised submission). I find that the authors have addressed appropriately the concerns that were previously raised, especially the reformulation of the study’s aims, the global organisation and the more modest tone used in the abstract. This revised version is much clearer, more precise, better organised and more readable than the previous one. I only have a few minor comments

The writing could still be improved, notably by shortening sentences, but the discourse is clear enough to be understood. Some English mistakes remain (I attempted at pointing them out in the specific comments).

Regarding the method section, great efforts have been made to provide more details on the analyses and the tests used, and to organise the analyses to mirror the results presentation. I think there is still one analysis description missing: I am guessing the results section entitled “Egg laying patterns within individuals” (L328-353) are from similar analyses to those described in the “intra-clutch egg-size variation” and “patterns of egg size variation in relation to laying order”, but this is not specified in the methods section. Adding one more subsection stating something like “we reproduced the analyses described in sections ‘…’, limiting them to recaptured females that had changed clutch size from 4 to 5 eggs between years” would do the job. Also, I think that, to ensure replicability and transparency, some more details could be provided, for example the models distribution when not using linear models (Poisson?).

As to the statistical features reported, I reckon it is a matter of opinion. I find slopes and SEs more informative as it provides with a direction and a strength of the differences. However, now that the statistics section is more detailed and much clearer than previously, I agree that F-values are inherent to the comparison process used in the analyses of this manuscript. As to the R² values, I have never used SAS, but is it not possible to manually calculate marginal and conditional R² for (G)LMMs, using Nakagawa and Schielzeth (2013 Methods in Ecology and Evolution) approximation? Just so that results are more consistently presented…

Specific comments (line numbers referring to the manuscript version with tracked changes):

Abstract

L22 and throughout: please harmonize between “intraclutch” and “intra-clutch”

L23: please add commas “to clutch size, and the relation between clutch size and female body condition,”

L25-26: I think the authors mean “within the laying sequence depending on clutch sizes in response to body condition”

L26: please reformulate “as such an adjustment could have adaptive implications on”

L28: “advantage for the hatchlings”

L28: “The analyses were first performed at the population level; and then at the individual level using data from”

L31-32: this part is a bit heavy, I suggest reformulating, for example as “(range: 1–6), 4- and 5-egg clutches constituted c.70% of all clutches, and taking turns in being the most represented clutch size.”

L34: “levels”

L39: if I understand correctly, the punctuation should be as follows “pre-laying body condition and clutch size, and the intraclutch”

L40: “pattern indicate that both clutch and egg size are actively”

L43: please nuance by replacing “indicates “with “suggests”

L44: “mechanism” (singular)

L45: “years where females are in suboptimal body condition”

Introduction

L61: “in a population”

L63: “of the first egg”; “efficiency of females”

L67-68: “number, to incubation strategy, to facilitation […] and to differential”

L72: “Although egg size”

L76: “which lay”

L83: by “equal”, do the authors mean “even” (as in ‘make equal’)?

L85: “life history” (no hyphen)

L93-94: “as well as an increase in reproductive success”

L102: “breeding populations, suggesting that eiders could”

L105: “if common eider females change egg size”

L109-110: “data at the population level, as well as the individual level”

Methods

L120: “held” (not holded)

L133-134: “”in cases where no egg-shell remains were found to disqualify the record)

L156: do you mean “hatched eggs” (instead of hatching eggs)?

L171: “To ensure”

L191: is there a difference between general linear model and generalized linear model (L203)? I suspect not, therefore please pick one. Also, if it is a GLM, please specify the response variable error distribution (I am assuming a Poisson one?)

L208: do you mean “Intraclutch” (instead of Interclutch)? Cf. L279

L210: is it a linear mixed model?

L211: “as a fixed effect and year as a random factor”

L215-216: “and year, and interactions between laying order and clutch size, and between year and laying order”

L218: “bypasses”

L220: “egg size could only be”

L221-222: “(e.g. the pattern for the first four eggs could only be tested in clutches”

L222-223: do you mean for each clutch size or that you conducted separate tests on first eggs, then on the first 2 eggs, then the first 3 eggs… (I assume the latter, therefore you might wann reformulate)

L238: “were fulfilled”

Results

L257: “with 4-egg clutches being more frequent in some years”

L261: “Clutch size was”

L263: “0.0001” (instead of 0,0001)

L266 & 268: “The majority of the recaptured females”; “size the following year”

L267 & 269: remove “had” in both instances

L270: “Among the 108 females that did” (no comma)

L272: “and 65 gained”

L274-275: do you mean p < 0.0001 (instead of >)? No capital letter is needed for “slope”

L296: there is one too many closing parenthesis

L301: “data were analysed” (data is plural)

L314: please remove “as indicated by the significant effect of laying order”

L334: with “marked”, do you mean “marginal”?

L338: please replace “evident” with “observed”

Discussion

L364-365: please remove “(although just insignificant)”. I understand you want to remind this point, so I suggest you replace “marginally” by “almost significantly”

L366: “females when they both increased”

L368: please replace “data” with “results”

L378: “laying date varied yearly? by only”

L389: “reflect” (data is plural)

L394: “with” instead of “within”, as I assume you mean females produce eggs according to a specific pattern

L397: “contemporary” is not too clear, I would use “associated pre-laying”

L413: do you mean “in terms of number of ducklings”?

L414: “duckling survival” to what?

L418: “in size with the two”

L422: “the likely associated fitness benefits.”; “waterfowls?” (I am not sure if it is a mistake or if fowl has an irregular plural, same L450)

L433: “Analysed in relation to”

L435: “show” (data is plural)

L441: “5- and 6-egg” (there are some hyphens missing every now and then, also L475, 493, 495)

L442: “in terms of simple size” is not clear, if you mean ‘regarding size’, I think that can be removed as the point is made by the rest of the sentence

L443: please remove “both”, or reformulate as it is unclear which two elements it refers to

L454: either ‘the highest energy investment” or “higher energy investment” (although I think you mean allocation, as you refer to an increase in egg size but not necessarily associated with a greater cost to the female?)

L456-457: “eggs is much higher” (refers to pre-incubation failure, singular) and “eggs is attributed” (refers to survival, singular)

L458: “start” (plural, refers to both nest attendance and incubation initiation)

L502: please replace “specific” with “recaptured”

L503: “when in poor condition”

L504: “we argue that” (no comma)

L508: “clutches, we hypothesise”

7. PLOS authors have the option to publish the peer review history of their article (what does this mean?). If published, this will include your full peer review and any attached files.

Reviewer #1: Yes: PAULINE TONI

---

## [Author Response · Author response to Decision Letter 1]

15 Jun 2020

6. Review Comments to the Author

Reviewer #1: This manuscript aims at testing how intra-clutch egg size varies with laying sequence depending on clutch size, and makes the link between such variation and maternal body condition, in common eiders. The authors use long-term, longitudinal data to answer these questions at both the population and the individual level. The study is of interest, as the results are sound and based on an impressively large dataset. The analyses at the population and individual levels bring important insights on egg size variation patterns. The paper offers an interesting discussion and interpretation of the results in the larger framework of female reproductive strategies.

This is the second time I review this manuscript (revised submission). I find that the authors have addressed appropriately the concerns that were previously raised, especially the reformulation of the study’s aims, the global organisation and the more modest tone used in the abstract. This revised version is much clearer, more precise, better organised and more readable than the previous one. I only have a few minor comments

The writing could still be improved, notably by shortening sentences, but the discourse is clear enough to be understood. Some English mistakes remain (I attempted at pointing them out in the specific comments).

Regarding the method section, great efforts have been made to provide more details on the analyses and the tests used, and to organise the analyses to mirror the results presentation. I think there is still one analysis description missing: I am guessing the results section entitled “Egg laying patterns within individuals” (L328-353) are from similar analyses to those described in the “intra-clutch egg-size variation” and “patterns of egg size variation in relation to laying order”, but this is not specified in the methods section. Adding one more subsection stating something like “we reproduced the analyses described in sections ‘…’, limiting them to recaptured females that had changed clutch size from 4 to 5 eggs between years” would do the job. Also, I think that, to ensure replicability and transparency, some more details could be provided, for example the models distribution when not using linear models (Poisson?).

The statistics for the section: ‘ Egg-laying patterns within individuals’ was not described in the previous version. When examining it again we realized that our test did not account for the paired design. We therefore reanalyzed the data using a paired t-test. This did not change the significance of the results. We apologize for overlooking this issue earlier.

As to the statistical features reported, I reckon it is a matter of opinion. I find slopes and SEs more informative as it provides with a direction and a strength of the differences. However, now that the statistics section is more detailed and much clearer than previously, I agree that F-values are inherent to the comparison process used in the analyses of this manuscript. As to the R² values, I have never used SAS, but is it not possible to manually calculate marginal and conditional R² for (G)LMMs, using Nakagawa and Schielzeth (2013 Methods in Ecology and Evolution) approximation? Just so that results are more consistently presented…

We do not use R2 estimates for interpreting our results. We doubt that addition of R2 to the generalized linear mixed models would add much information. In addition, the R2 value is not simple to estimate in mixed and generalized mixed models, and the interpretation across statistical models is far from trivial. 

Specific comments (line numbers referring to the manuscript version with tracked changes):

Abstract

L22 and throughout: please harmonize between “intraclutch” and “intra-clutch” Corrected

L23: please add commas “to clutch size, and the relation between clutch size and female body condition,” Corrected

L25-26: I think the authors mean “within the laying sequence depending on clutch sizes in response to body condition” Corrected

L26: please reformulate “as such an adjustment could have adaptive implications on” Corrected

L28: “advantage for the hatchlings” Corrected

L28: “The analyses were first performed at the population level; and then at the individual level using data from” Corrected

L31-32: this part is a bit heavy, I suggest reformulating, for example as “(range: 1–6), 4- and 5-egg clutches constituted c.70% of all clutches, and taking turns in being the most represented clutch size.” Corrected

L34: “levels” Corrected

L39: if I understand correctly, the punctuation should be as follows “pre-laying body condition and clutch size, and the intraclutch” Corrected

L40: “pattern indicate that both clutch and egg size are actively” Corrected

L43: please nuance by replacing “indicates “with “suggests” Corrected

L44: “mechanism” (singular) Corrected

L45: “years where females are in suboptimal body condition” Corrected

Introduction

L61: “in a population” Rephrased to “Different eider populations” (as this is shown in several different populations.

L63: “of the first egg”; “efficiency of females” Corrected

L67-68: “number, to incubation strategy, to facilitation […] and to differential” Corrected

L72: “Although egg size” Corrected

L76: “which lay” Corrected

L83: by “equal”, do the authors mean “even” (as in ‘make equal’)? Yes, new phrasing: “make equal”.

L85: “life history” (no hyphen) Corrected

L93-94: “as well as an increase in reproductive success” Corrected

L102: “breeding populations, suggesting that eiders could” Corrected, but the word “could” is not included, as this makes the sentence less precise. 

L105: “if common eider females change egg size” Corrected

L109-110: “data at the population level, as well as the individual level” Corrected

Methods

L120: “held” (not holded) Corrected

L133-134: “”in cases where no egg-shell remains were found to disqualify the record) Corrected

L156: do you mean “hatched eggs” (instead of hatching eggs)? Hatching nests refer to nests where ducklings were not hatched, but where eggs were in the process of hatching. Hatched eggs is not a precise description as such nests will be similar to nests where females were captured with ducklings. 

L171: “To ensure” Corrected

L191: is there a difference between general linear model and generalized linear model (L203)? I suspect not, therefore please pick one. Also, if it is a GLM, please specify the response variable error distribution (I am assuming a Poisson one?) General linear models assume normal distribution whereas generalized linear models assumes poisson or other non-normal distribution. We have specified the use of non-normal distributions (Generalized linear models) in the statistic paragraph.

L208: do you mean “Intraclutch” (instead of Interclutch)? Cf. L279 As these analyses consider both within and between clutch variation, we have changed the heading to “Inter- and intraclutch egg-size variation in both the Statistical and Result sections. 

L210: is it a linear mixed model? As laying order was coded as a categorical variable and as such does not model a linear relation we think that mixed model is the most appropriate term. 

L211: “as a fixed effect and year as a random factor” Corrected

L215-216: “and year, and interactions between laying order and clutch size, and between year and laying order” Corrected

L218: “bypasses” Corrected

L220: “egg size could only be” Corrected

L221-222: “(e.g. the pattern for the first four eggs could only be tested in clutches” Corrected 

L222-223: do you mean for each clutch size or that you conducted separate tests on first eggs, then on the first 2 eggs, then the first 3 eggs… (I assume the latter, therefore you might wann reformulate) No. There needs to be at least 2 eggs in a clutch for testing laying order. We have rephrased the sentence.

L238: “were fulfilled” Corrected

Results

L257: “with 4-egg clutches being more frequent in some years” Corrected

L261: “Clutch size was” Corrected

L263: “0.0001” (instead of 0,0001) Corrected

L266 & 268: “The majority of the recaptured females”; “size the following year” First part is changed, but the second part (size the following year) is kept, as clutch size change did not in all instances take place from one year to the next. Some birds were not captured every year, but every second year. 

L267 & 269: remove “had” in both instances Corrected

L270: “Among the 108 females that did” (no comma) Corrected

L272: “and 65 gained” Corrected

L274-275: do you mean p < 0.0001 (instead of >)? No capital letter is needed for “slope” Yes and Corrected

L296: there is one too many closing parenthesis Corrected

L301: “data were analysed” (data is plural) Corrected

L314: please remove “as indicated by the significant effect of laying order” Corrected

L334: with “marked”, do you mean “marginal”? No. With the t-test applied the significance is marked

L338: please replace “evident” with “observed” Corrected

Discussion

L364-365: please remove “(although just insignificant)”. I understand you want to remind this point, so I suggest you replace “marginally” by “almost significantly” Corrected

L366: “females when they both increased” Corrected

L368: please replace “data” with “results” Corrected

L378: “laying date varied yearly? by only” We have changed “colony” with “annual” to point out that yearly laying dates vary by only 5 days 

L389: “reflect” (data is plural) Corrected

L394: “with” instead of “within”, as I assume you mean females produce eggs according to a specific pattern Corrected

L397: “contemporary” is not too clear, I would use “associated pre-laying” Corrected

L413: do you mean “in terms of number of ducklings”? YES- Corrected

L414: “duckling survival” to what? We have specified this: “higher duckling survival is the referred consequences.

L418: “in size with the two” Corrected

L422: “the likely associated fitness benefits.”; “waterfowls?” (I am not sure if it is a mistake or if fowl has an irregular plural, same L450) First part corrected. Second part: new phrasing is “waterfowl species”, which should be the correct (changed both places).

 L433: “Analysed in relation to” Corrected

L435: “show” (data is plural) Corrected

L441: “5- and 6-egg” (there are some hyphens missing every now and then, also L475, 493, 495) Corrected

L442: “in terms of simple size” is not clear, if you mean ‘regarding size’, I think that can be removed as the point is made by the rest of the sentence Corrected - deleted

L443: please remove “both”, or reformulate as it is unclear which two elements it refers to Corrected

L454: either ‘the highest energy investment” or “higher energy investment” (although I think you mean allocation, as you refer to an increase in egg size but not necessarily associated with a greater cost to the female?) First part corrected. It should be clear that it is an “allocation” from the previous sentence. 

L456-457: “eggs is much higher” (refers to pre-incubation failure, singular) and “eggs is attributed” (refers to survival, singular) Corrected

L458: “start” (plural, refers to both nest attendance and incubation initiation) Corrected

L502: please replace “specific” with “recaptured” Corrected

L503: “when in poor condition” Corrected

L504: “we argue that” (no comma) Corrected

L508: “clutches, we hypothesise” Corrected

---

## [Editor Report · Decision Letter 2]

9 Jul 2020

Condition dependent strategies of egg size variation in the Common Eider Somateria mollissima

PONE-D-19-33049R2

Dear Dr. Christensen,

We’re pleased to inform you that your manuscript has been judged scientifically suitable for publication and will be formally accepted for publication once it meets all outstanding technical requirements.

Kind regards,

Julien Martin

Academic Editor

PLOS ONE

---

## [Editor Report · Acceptance letter]

13 Jul 2020

PONE-D-19-33049R2 

Condition dependent strategies of egg size variation in the Common Eider Somateria mollissima 

Dear Dr. Christensen:

I'm pleased to inform you that your manuscript has been deemed suitable for publication in PLOS ONE. Congratulations! Your manuscript is now with our production department. 

Kind regards, 

on behalf of

Dr. Julien Martin 

Academic Editor

PLOS ONE